# Optimizing Hydroxyurea Treatment for Sickle Cell Disease Patients: The Pharmacokinetic Approach

**DOI:** 10.3390/jcm8101701

**Published:** 2019-10-16

**Authors:** Charlotte Nazon, Amelia-Naomi Sabo, Guillaume Becker, Jean-Marc Lessinger, Véronique Kemmel, Catherine Paillard

**Affiliations:** 1Hôpitaux Universitaires de Strasbourg, Centre de compétence pour les maladies constitutionnelles du globule rouge et de l’érythropoïèse, Service d’hématologie oncologie pédiatrique, Avenue Molière, 67200 Strasbourg, France; charlotte.nazon@chru-strasbourg.fr; 2Laboratoire de Pharmacologie et Toxicologie Neurocardiovasculaire, Faculté de Médecine, 11 rue Humann, 67085 Strasbourg, France; amelia-naomi.sabo@chru-strasbourg.fr (A.-N.S.); Jean-Marc.lessinger@chru-strasbourg.fr (J.-M.L.); 3Hôpitaux Universitaires de Strasbourg, Hôpital de Hautepierre, Laboratoire de Biochimie et Biologie Moléculaire, Avenue Molière, 67200 Strasbourg, France; guillaume.becker@chru-strasbourg.fr; 4Hôpitaux Universitaires de Strasbourg, Service de la Pharmacie, Avenue Molière, 67200 Strasbourg, France; 5Laboratoire d’ImmunoRhumatologie Moléculaire, INSERM UMR_S 1109, LabEx Transplantex, Fédération de Médecine Translationnelle de Strasbourg, 4 rue Kirschleger, 67085 Strasbourg Cedex, France

**Keywords:** sickle cell disease, sickle cell anemia, hydroxyurea, pharmacokinetics

## Abstract

Background: Hydroxyurea (HU) is a FDA- and EMA-approved drug that earned an important place in the treatment of patients with severe sickle cell anemia (SCA) by showing its efficacy in many studies. This medication is still underused due to fears of physicians and families and must be optimized. Methods: We analyzed our population and identified HU pharmacokinetic (PK) parameters in order to adapt treatment in the future. Working with a pediatric population, we searched for the most indicative sampling time to reduce the number of samples needed. Results: Nine children treated by HU for severe SCA were included for this PK study. HU quantification was made using a validated gas chromatography/mass spectrometry (GC/MS) method. Biological parameters (of effectiveness and compliance) and clinical data were collected. None of the nine children reached the therapeutic target defined by Dong et al. as an area under the curve (AUC) = 115 h.mg/L; four patients were suspected to be non-compliant. Only two patients had an HbF over 20%. The 2 h sample was predictive of the medication exposure (*r*^2^ = 0.887). Conclusions: It is urgent to be more efficient in the treatment of SCA, and pharmacokinetics can be an important asset in SCA patients.

## 1. Introduction

Sickle cell anemia (SCA) is one of the most common inherited diseases in the world. It affects more than 300,000 infants born annually worldwide, and the epidemiologic projections show a growing tendency for the years to come (30% increase by 2050) [1]. It is an autosomal recessive disease affecting the red blood cells due to a point missense mutation on the hemoglobin beta chain. This mutation leads to a polymerization of the hemoglobin (Hb), which causes an increased density, dehydration, and deformation of red blood cells, forming the sickle cells [2].

Clinical manifestations of SCA include hemolytic anemia, vaso-occlusive crisis (VOC), and bacterial susceptibility and can have an impact on many organs [3,4,5,6,7,8,9].

Hydroxyurea (HU) is a FDA- and EMA-approved drug that earned an important place in the treatment of patients with severe SCA. It has shown its efficacy in multiple studies by reducing the morbi-mortality and frequency of VOC, transfusions, and hospitalizations for those patients [10,11,12,13,14,15,16]. Furthermore, HU treatment is associated with improvement in hemoglobin concentration illustrated by increasing mean corpuscular volume (MCV) and Hemoglobin F (HbF) levels [17]. One way to adjust HU dose, mostly used in American centers, is to introduce HU at 15–20 mg/kg/day and to make a dose escalation until a mild myelosuppression tolerated by the patient is obtained, which indicates that the maximum tolerated dose (MTD) has been reached [18]. The dose escalation of HU depends on three hematological parameters: the neutrophil count (1.5–3 G/L), the reticulocyte count (100–200 G/L), and the platelet count (>80 G/L) [17]. When the MTD is reached, the risk/benefit balance is optimal for the patient [19]. Despite the fact that the escalation to MTD has proven to be the best way to dose HU, many European centers use a fixed-dose strategy (20 mg/kg/day).

The dose escalation method presents some drawbacks: it requires frequent outpatient visits and laboratory tests and usually takes 6 to 12 months; the treatment is usually suboptimal during that time [20,21,22]. The fact that there is a period during which the treatment is not providing any clinical improvement, and that frequent laboratory tests and medical consultations are needed, does not help patient adherence, which is already low in this population [23,24]. It has been highlighted by Brandow et al., who showed that the patients who refused HU gave the following reasons: fear of cancer and other side effects in the majority followed by not wanting to take medication, not wanting to have required laboratory monitoring, or not thinking the medication would work [23]. Regarding toxicities, even if we now know that the myelosuppression is transient [18,25], that the azoospermia caused by HU could be reversible [26], and that there is no evidence it has a genotoxic or a leukemogenic effect [15], the reticence among physicians and families is still strong, leading to an underuse of this treatment [23]. Studies showed that the escalation to MTD did not add any toxicities [27]. Organ damage begins early in life, worsens over time, and is irreversible. Accordingly, early optimal treatment in young patients who have not yet developed serious or irreversible organ damage is a necessity.

Another difficulty remains in pharmacokinetic variations of HU. Indeed, for a same dose of HU, drug exposure may vary five times in adults and three times in children [25]. There are major inter-individual variations regarding absorption, profiles, distribution, and clearance of HU [28]. Logically, HU is not efficacious at the same dosage for everyone: In a study of Dong et al., posology at MTD ranges from 14.2 to 35.5 mg/kg/day [21]. It makes standard patient care impossible.

For these reasons, a personalized dose optimization process that can rapidly identify MTD for individual SCA patients is highly desirable. The main goal of this study was to analyze our population and identify the pharmacokinetic parameters of HU to be able to adapt and optimize their treatment in the future. By doing so, we intend to have a model we can use to adapt the dose of HU and reach the MTD for an earlier clinical benefit for our patients more quickly. Working with a pediatric population, we wanted to find the most indicative sampling time to reduce the number of blood collections needed.

## 2. Experimental Section

### 2.1. Patients

We included, prospectively, all the patients with SCA of <20 years of age, treated by HU attending our hospital (Hôpitaux Universitaires de Strasbourg, France), for a follow-up consultation between February and May 2018. More than 100 patients with SCA (SS or SC) are followed in our hospital, including 27 treated by HU. In our hospital, HU is introduced at 15 mg/kg/day and normally increased to reach MTD following hematological criteria: the neutrophil count (1.5–3 G/L), the reticulocyte count (100–200 G/L), and the platelet count (>80 G/L). Although, most of the time, if there is no clinical manifestation of SCA, no dose adjustment is done.

All patients and/or parents/guardians provided written informed consent before enrollment in the study. This study received approval from the Institutional review board of Strasbourg University Hospital (DRCI 2018-project n°6112) and the French data protection authority (CNIL-n°2215437).

### 2.2. Study Design

Plasma samples were collected at the following times: pre-dose and at 10 min, 20 min, 1 h, 2 h, 4 h, and 6 h after oral HU administration at the patient’s usual dose. Whole blood samples were transported and/or stored at 2–8 °C for a maximum of four h before centrifugation, and then aliquoted plasma was rapidly frozen at −20 °C.

Demographic information and standard laboratory parameters were collected: Neutrophil, reticulocyte, and platelet counts were measured in turn to identify the individual tolerability of the treatment, hemoglobin, MCV, and the percentage of HbF to monitor the clinical efficacy of HU.

We evaluated the adherence of our patients by asking them about the frequency of missed doses over the past six months. Genotypic data were collected, looking for associated alpha thalassemia and haplotype.

### 2.3. GC-MS

The dosage of HU was made by our hospital’s Biochemistry and Molecular Biology Laboratory using a gas chromatography coupled with mass spectrometry (GC-MS) technique (version 2.0, CHU Strasbourg, Strasbourg, France), which can be used routinely. A Thermo Scientific™ ISQ Series (Thermo Fisher Scientific, Courtaboeuf, France) was used for the analysis. Chromatography was performed on a low polarity RTX-5-MS (30 m × 0.25 mm × 0.25 µm) capillary column, and helium was the carrier gas. The total analysis time of a sample was 13 min. Data were obtained from the Thermo Xcalibur^©^ software (^©^ 2012 Thermo Fisher Scientific Inc., version 3.1, Courtaboeuf, France).

It is a sensitive and specific method validated by our local laboratory that is feasible with a low quantity of plasma (50 μL). The measurement of this method ranges from 0.79 to 100 mg/L with a detection limit of 0.28 mg/L.

### 2.4. Modelling and Statistical Analysis

The plasma concentration–time data was analyzed by a non-compartmental method to obtain a concentration–time curve. The pharmacokinetics’ parameters were defined for each patient: the maximum plasma concentration (C_max_), the time to reach the C_max_ (T_max_), and the area under the curve (AUC). The C_max_ were identified by a graphical analysis.

The mean, standard deviation, and median were calculated using Microsoft Excel. The simulated AUC_0–6h_ were calculated by the linear log trapezoidal rule (version 6, GraphPad Prism, San Diego, CA, USA).

### 2.5. Optimal Sampling

Exploration of the relationship between HU concentration–time and exposition was made. The significant linear correlations were defined by the determination coefficient *r*^2^ > 0.5. AUC was tested by a non-compartmental method. Data analysis was made using the GraphPad Prism^®^ program (version 6, GraphPad Prism, San Diego, CA, USA).

## 3. Results

### 3.1. Characteristics of Patients

Nine patients were included, and their characteristics are shown in Table 1. These patients were on HU treatment for multiple VOC or acute chest syndrome (ACS). Most of them were treated by HU for more than four years, and their daily doses ranged from 12.9 to 24.6 mg/kg/day.

### 3.2. Biological Parameters and Self-Reported Compliance

Biological parameters are presented in Table 2. Seven patients had an HbF lower than 20%. Four children had a normal or low MCV. None of the patients reached the myelosuppression as defined earlier as a sign of MTD of HU. None of the patients showed a major hematological toxicity. However, Patient 8 had the lowest neutrophil count, middle MCV, and a low percentage of HbF. We evaluated the adherence of our patients by asking them about the frequency of missed doses over the past six months. We defined low compliance level as one missed-dose per week or more (*n* = 2), medium compliance level as one to three missed-doses per month (*n* = 4), and high compliance level as less than one dose-missed a month (Table 2) (*n* = 3). The genotypic profile was available only for Patient 8, who is not a carrier of alpha thalassemia and has a Benin/Benin (BEN/BEN) haplotype.

### 3.3. Pharmacokinetic Parameters

Principal pharmacokinetic parameters are presented in Table 3 and mean HU concentration-time are represented in Figure 1. The AUC ranged from 43.3 to 113.5 h.mg/L with a median of 75.1 h.mg/L. None of the nine children reached 115 h.mg/L, which was the target-AUC in a study by Dong et al. [21]. Six of them had an AUC less than 100 h.mg/L. However, Patient 2 presented with an AUC of 113.5 h.mg/L and had one of the highest HbF percentages (20.5 %). Patient 6, who had the highest HbF (=23.7%), had an AUC of 74.0 h.mg/L, showing a non-optimal response. Every child except one had a time to reach the C_max_ (T_max_) between 1 and 2.5 h.

### 3.4. Optimal Sampling

Using a non-compartmental method, a more or less significant correlation appears between AUC and the concentrations measured at the same time of sampling. As shown in Figure 2a, the most significant correlation was obtained at the 2-h sampling time (*r*^2^ = 0.8775). A less significant correlation (Figure 2b) was found for the 4-h concentrations (*r*^2^ = 0.6058). The best correlation was found for the 2-h samples, which could be sufficient to predict the patient AUC (Table 4).

## 4. Discussion

In this study, we analyzed the pharmacokinetic parameters of a population of nine children in order to appreciate the possibility of medical care improvement using PK analysis. This opportunity could allow us to adapt their treatment in a more efficient way.

The GC/MS method had the specificity and sensitivity required for the therapeutic follow-up of HU. It was linear from 0.79 to 100 mg/L and had a detection limit of 0.28 mg/L. All these qualities are compatible with the plasma concentrations found in adults and infants [25,29]. Even if many HU dosage techniques have been published, only those using mass spectrometry as a detector are specific enough not to interfere with endogenous compounds and allow low concentration quantification [30,31,32].

Ware et al. described two phenotypic absorption profiles: “Fast” (defined as C_max_ reached at 15 or 30 min) and “Slow” (C_max_ reached at 60 or 120 min) [28]. In our study, we also observed a “fast” and “slow” absorption profile but with C_max_ reached before two hours and after two hours, respectively. This indicates a slower absorption profile, but six out nine were with “fast” profiles and only three with a “slow” one. Ware et al. described the same proportion between slow and fast profiles despite the time-shift.

We highlighted that our patient care is sub-optimal. None of the patients had reached the myelosuppression that was defined earlier as the MTD. Dong et al., using a Bayesian analysis approach, published that 115 h.mg/L could be chosen as the target AUC to reach at the HU initiation [21]. Regarding the pharmacokinetic parameters in our study, none of the patients reached the MTD. Our AUC results were in accordance with observations by Dong et al. before MTD was reached. Moreover, our low AUC results were in accordance with the first administration of HU found in Ref. [21,33]. The highest HbF was 23.7%, while McGann et al. showed that the average HbF of their population was 33.3 ± 9.1% after 12 months of treatment at MTD [33].

The first explanation for our patients not reaching MTD resides in the dose. The administrated doses in our population ranged from 12.9 to 24.6 mg/kg/day, while in the study by Dong et al., when the MTD was reached, doses were between 14.2 and 35.5 mg/kg/day. Despite their long treatment durations (between 11.2 and 138.8 months), we could see that the dose escalation to reach MTD was not done properly for these patients. It is clear that most of the patients did not have the appropriate dose.

Secondly, four out of nine children were suspected as non-compliant due to their low or normal MCV; however, using laboratory parameters to assess HU adherence can be misleading since the increased MCV and HbF is not universal [23]. Moreover, the genotyping of our population was performed for only one patient, so the association with alpha thalassemia explaining a low MCV cannot be ruled out. MCV prior to HU initiation was not available for all the patients because the treatment was initiated in another country or medical center. Another limitation of our study is the use of patients in order to assess compliance. This method is often unreliable, but some patients admitted their non-adherence. The use of a combination of methods (laboratory parameters, pill counts, logbook) is necessary. Our results are yet consistent with the studies showing the physicians reticence to increase HU dosages and that the adherence of SCA patients to HU is average [23,24].

Another piece of information that has proven its importance is that of the beta haplotypes. Unlike the United States, which has a high predominance of BEN haplotypes, there is a larger proportion of patients with Central African Republic (CAR) haplotypes. It has been shown by Bernaudin et al. that BEN/BEN patients have a better response to HU than CAR/CAR patients [34]. Only one result of the haplotype was available, and the patient had a BEN/BEN haplotype. The rest of the population must be explored knowing the influence on the course of the disease it has.

The one pitfall of this study is, of course, the sample size, which was mainly due to the difficulty of implementation. In fact, the most restrictive aspect of this method was the number of blood samples needed. Working with a pediatric population, we wanted to find an optimal sampling time that could predict the exposure to the medication. The best correlation between AUC and HU concentrations was found at two hours. This result is consistent with the literature [29,35] and would make PK analysis easier to apply routinely.

Furthermore, a recent study (TREAT, ClinicalTrials.gov NCT02286154) demonstrated that the PK guided dose strategy, with a target set at 115 h.mg/L, was more efficient and without excess hematologic toxicities than classical dose escalation based on hematological parameters [33].

This pharmacokinetic approach, offering no additional toxicities and optimizing the dose of HU more efficiently and quickly, is an important asset in SCA patient care. Reaching MTD will help compliance by helping the patients see the quick benefits of taking this medication. This is all the more important given the necessity to be efficient before the occurrence of irreversible organ damage.

## 5. Conclusions

It is urgent to be more efficient in the treatment of SCA patients. Specifically, risks of HU must be compared with the risks of untreated SCA; the natural history of clinically severe SCA is well known with a poor prognosis [9,36,37,38]. We need to stop underusing HU, which has proven its benefits for years. We have to embrace the concept of dose escalation, and we need to find ways to be optimal. The PK approach could be one such way.

## Figures and Tables

**Figure 1 jcm-08-01701-f001:**
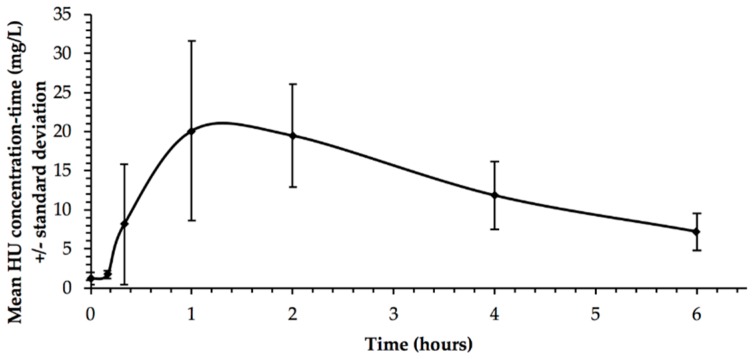
Mean HU (±standard deviation) concentration–time plot (*n* = 9 patients).

**Figure 2 jcm-08-01701-f002:**
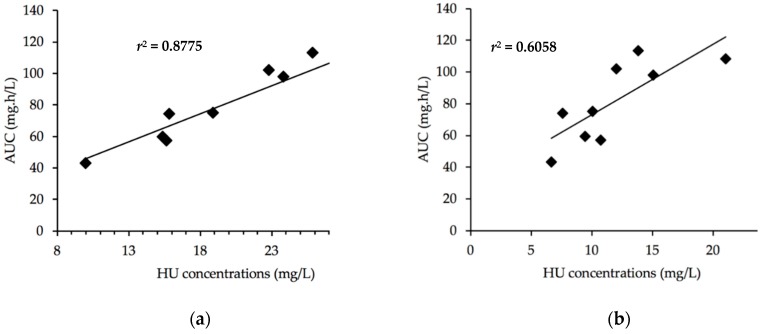
(**a**) Relation between area under the curve (AUC) and HU concentration-time for 2-h samples; y = 3.5701x + 11.727. (**b**) Relation between AUC and HU concentration-time for 4-h samples; y = 4.4637x + 28.402.

**Table 1 jcm-08-01701-t001:** Patient characteristics.

Demographic Characteristics	
Sex ratio M/F	0.8 (4/5)
Age	
Mean ± standard deviation	14.4 (±3.7)
Median	16.5
Weight	
Mean ± standard deviation	49.9 (±20.5)
Median	49.1
Background (Number of Patients and Percentage)	
Cholecystectomy	3 (33%)
Stroke	1 (11.1%)
Abnormal Transcranial doppler episode	1 (11.1%)
Osteonecrosis	2 (22.2%)
Retinopathy	1 (11.1%)
Splenic Sequestration	0 (0%)
Pulmonary Hypertension	1 (11.1%)
Cardiac Events	1 (11.1%)
Kidney Failure	0 (0%)
Events per Year: 2016–2018 Period	
Transfusion/Year	
Mean ± Standard Deviation	0.8
Median	0.3 (0–2)
Hospitalization/Year	
Mean ± standard deviation	1.4
Median (range)	0.6 (0–5.0)
VOC/Year	
Mean ± Standard Deviation	1.6
Median (range)	1 (0–5.6)
ACS	
Number of Patients > 1 ACS	4 (45%)
HU	
Dose (mg/kg/day)	
Mean ± Standard Deviation	19.0 (±4.0)
Median (range)	20.4 (12.9–24.6)
Time since Introduction of HU (Months)	
Mean ± Standard Deviation	63.5 (±44.6)
Median (Range)	58.8 (11.2–138.8)
Age at Introduction (Year)	
Mean ± Standard Deviation	8.5 (±4.4)
Median (Range)	6.0 (4.0–16.0)

HU: Hydroxyurea; VOC: vaso-occlusive crisis; ACS: acute chest syndrome. M: male; F: female.

**Table 2 jcm-08-01701-t002:** Self-reported compliance, biological parameters, and HU intake characteristics of the nine children on HU.

Patient	Self-Reported Compliance	Hb (g/dL)	MCV (fL)		Retic. (G/L)	PNN (G/L)	Platelets (G/L)	HbF(%)		Dose HU (mg/kg/day)
			Pre-HU	Post-HU				Pre-HU	Post-HU	
1	poor	8.6	81.2	87.5	284.0	6.0	427	3.3	7.0	17.1
2	good	9.0	86.3	111.9	161.2	4.9	282	8.8	20.5	20.9
3	poor	7.7	N/A	70.2	192.7	11.5	279	N/A	3.0	20.4
4	medium	7.1	83.0	95.8	242.2	14.2	81	4.8	7.9	21.4
5	medium	8.3	79.0	78.1	286.7	8.0	539	N/A	1.4	21.4
6	good	9.1	N/A	94.9	185.0	5.9	232	N/A	23.7	18.9
7	medium	7.8	88.0	80.4	325.2	9.2	589	5.7	7.4	24.6
8	medium	7.6	78.4	95.0	177.1	3.2	290	2.5	5.6	13.0
9	good	7.6	N/A	97.8	164.7	4.9	465	N/A	14.5	12.9
Mean ± SD		8.1 ± 0.7		90.2 ± 12.5	224.3 ± 61.5	7.5 ± 3.6	354 ± 163		10.1 ± 7.8	19.0 ± 4.0
Median		7.8		94.9	192.7	6.0	290.0		7.4	20.4

Pre-HU: parameters before HU initiation; Post-HU: parameters after HU initiation.

**Table 3 jcm-08-01701-t003:** Pharmacokinetics parameters of the nine children on HU.

Patient	Treatment Duration (Months)	Dose (mg/kg/day)	C_max_ (mg/L)	T_max_ (hours)	AUC (h.mg/L)
1	14.2	17.1	24.0	1.33	75.1
2	121.6	20.9	33.9	1.11	113.5
3	27.3	20.4	14.9	2.00	59.5
4	58.8	21.4	15.2	2.44	57.3
5	138.8	21.4	37.5	1.11	102.0
6	51.6	18.9	25.8	0.66	74.0
7	70.1	24.6	31.0	2.44	108.2
8	11.2	13.0	10.8	1.33	43.3
9	77.8	12.9	24.0	1.33	98.2
Mean ± Standard Deviation	63.5 (±44.6)	19.0 (±4.0)	24.1 (±9.1)	1.5 (±0.6)	81.3 (±25.2)
Median	58.8	20.4	24.0	1.33	75.1

**Table 4 jcm-08-01701-t004:** Determination coefficient *r*^2^ for the different sampling times.

Sampling Time	Determination Coefficient *r*^2^
0	0.0314
10 min	0.0146
20 min	0.0947
1 h	0.3415
2 h	0.8775
4 h	0.6058
6 h	0.4813

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
