# Peer review of "Optimizing Hydroxyurea Treatment for Sickle Cell Disease Patients: The Pharmacokinetic Approach"

_jcm, 2019, doi:10.3390/jcm8101701_

Round 1
Reviewer 1 Report
Thank you for submitting this interesting paper. I think a paragraph on study limitations e.g. sample size would improve the quality of this paper.
I have included minor English language edits on the attached. I hope this is useful

Author Response
Dear Editor and reviewer,
The modifications were indicated by colors to highlight changes. We would like to thank you for your time in evaluating our work and for your constructive comments that helped us to improve the quality of our manuscript.
We totally agree that the major limitation of our study is the sample size which is mainly due to the difficulty of implementation. We elaborated in the article the limitations of our study : sample size, the relability of self-reported interview.
Thank you for your time,
Best regards,

Reviewer 2 Report
In this manuscript, the authors describe an attempt to use a pharmacokinetics-based approach to select hydroxyurea doses for a small cohort of 9 children with sickle cell anemia. The PK data are obtained for children taking 15-20 mg/kg/day and have been described before, but it is an admirable effort by the authors to work towards an individualized dosing strategy. The optimization of hydroxyurea dosing is important and the message of the manuscript (nicely stated in the Conclusions) is important. Specific details below may help to improve the quality of the manuscript:
The written language is a bit difficult to read in parts. It would be helpful for a native English speaker to revise the manuscript to improve overall readability. Abstract (and text line 46): I would suggest removing the word “chemotherapeutic” as a descriptive term for hydroxyurea as this is part of the problems with its negative connotation. The definition of chemotherapy is “a drug used to treat cancer.” Though it has been used to treat cancer and is thus a “chemotherapeutic” drug, it should not be described this way when used as a treatment for sickle cell disease. We should try to avoid this word in the sickle cell literature as possible. Abstract (and text): is there evidence beyond the labs to suggest that the patients were not compliant? It is possible that laboratory parameters were not reached due to inadequate dose. Abstract (and text): The authors refer to the AUC target of 115 mg*h/L by Dong et al. This target was based upon patients who had dose escalated to MTD. The doses in this cohort are not escalated to MTD and it would not be expected that they would reach this AUC “target.” The PK data is consistent with the “baseline” PK data previously published using a dose of 20 mg/kg/day. The authors should clarify what the 115 mg*h/L target is based upon and why their patients do not reach this. The term “therapeutic drug monitoring” may not be the best term. The half-life of hydroxyurea is short and the drug is cleared within 12-24 hours. There is no accumulation of drug and there is no great benefit to measuring hydroxyurea levels to monitor drug response similar to how we may be able to do with anti-epileptics or antibiotics. The authors should consider revising this term. Intro: The authors state that “usually the dose is started at 15-20 mg/kg/day and is escalated to MTD.” While this is the preferred and seemingly the best way to dose hydroxyurea it is definitely not routine in many practices. Many providers start at a dose of 15-20 mg/kg/day and do not escalate the dose if there is clinical benefit. This is the typical dosing strategy in many European centers. The authors should discuss that dose escalation is not universal (but perhaps should be) and should speak to the dosing strategy in their center. Is the dose escalated? What is the starting dose? What are the laboratory criteria to escalate the dose if dose escalation is performed? The paragraph about adherence (beginning at line 64) is important but does not flow well from the previous paragraph. Prior to the “Brandow et al..” sentence, the concept of medication adherence should be addressed. In the Experimental Section, the authors should provide a background of their clinic and hydroxyurea guidelines. How big is the sickle cell population? The authors selected 9 patients but how many are in the entire sickle cell population and how many are on hydroxyurea? How were these 9 patients selected? As above, information about the standard use and dose strategies for hydroxyurea in this clinic should be described. The laboratory results would be more easily interpreted if “baseline” laboratory values before hydroxyurea were available. Though the authors suspect poor adherence, it is possible that the dose is just inadequate. Noting a change in MCV, ANC, Hb, and HbF are more indicative of medication adherence and response than a single cross-sectional value while on therapy. For example one patient may achieve HbF of 25% with a dose of 25 mg/kg but another may only have HbF of 10% on a similar dose, though both are adherent. Why did the authors use AUC (0-6h) instead of AUC (to infinity)? Table 1 is difficult to read. Reformatating would be helpful For the “Gravity” section of Table 1, are these numbers (transfusions, hospitalizations, ACS) events that have occurred in a lifetime? It may be more helpful to describe “per year” or count events within the last 1-2 years or something that may standardize for age. For the “HU” section of Table 1, it would be helpful to know what the age of hydroxyurea start was if this data is possible. Line 140-142, when the authors state “More curiously, patient 6…had a satisfying HbF% response..had an AUC of 74 mg*h/L.” Although 23.7% HbF is good, the data from Dong et al. suggest that if this dose is optimized (towards AUC of 115), the HbF% may be even greater. The authors should consider redefining what a “good” response is and think more along the lines of “optimal” response (often HbF>>20%). Discussion, line 164, it is not clear what the authors mean that the GC/MS method had “better specificity and sensitivity for the therapeutic follow-up of HU.” What is meantb y “better?” Discussion, line 174, the authors state that the children did not reach MTD “despite their long treatment durations,” but the concept of MTD requires dose escalation. If the dose is not adjusted, the dose will not become the MTD with time if it is not changed or increased. Do the clinical providers attempt to escalate the dose? Discussing the dosing strategy would be important to include. Particularly in Europe, the concept of dose escalation is less embraced. The authors have the chance to encourage a more aggressive dosing strategy here in this manuscript. The authors discuss that PK could be used to evaluate compliance, but this is not necessarily true. The half-life of hydroxyurea is short and hydroxyurea is cleared by the next day. So if you measure hydroxyurea levels, you really are only seeing whether the patient has taken a dose that day. The measurement of PK parameters and AUC would be useful to document that the dose is not near the target of 115 mg*h/L, but dose escalation can also be done simply by looking at the Complete Blood Count.
Author Response
Dear Editor and reviewer,
The modifications were indicated by colours to highlight changes.
Please find below a description of specific changes, points-to-point, replying to reviewers' comments. We reply to all the comments raised by the reviewer(s)
1) First, we made some changes to improve the writing and we hope it will be more understandable.
2) We removed the word « chemotherapeutic-drug ». You are right to say that it gave a negative connotation to this treatment and induce some fear to physicians and patients.
3) For the compliance, we asked the 9 nine patients about their adherence at the time of the study. We thought the patient interview was not reliable enough but some of them admitted their non-adherence. We add it to the article.
4) We believe our patients did not reach MTD for two reasons: first, the doses were too low as the escalation to MTD on hematological criteria was not properly made and second, the average compliance. We agree to say that compliance isn’t the only explanation and that the dose cannot the same for every child, we insisted on that
5) You are right to say that the therapeutic drug monitoring is not the best choice of term, we removed it.
6) It is true that in European centers, a lot of physicians use fixed-dose treatment for HU and the patient care can be very different depending on the habits of the medical staff. Therefore, we insisted on that notion.
In our hospital, HU is introduced at 15mg/kg/d and increased following hematological parameters, but it is true that the dose is often not raise to MTD due to reticence and acceptance of few clinical manifestations. That is a practice that we would like to change in our hospital and what we believe to be the case in many French centers. In our hospital we have more than 100 patients with SCA (SC or SS) including 27 who are treated by HU.
7) Baseline laboratory are now reported in the article. However, we don’t have those data for all the patients. The introduction of HU was often made in another country or in another hospital.
8) The gravity section of Table 1 has been redone: you can now find the mean and median transfusion, VOC, hospitalization per year on a 2 years period. We also add the age at HU introduction.
10) We insisted along the article on the notion of « optimal » response more than « good » response as you suggested.
11) The GC/MS method had the specificity and sensitivity required for the therapeutic follow-up of HU. "Better" was not the right term we remove it.
12) We meant by the « long duration of treatment » that during that time the treatment has been suboptimal and that the dose could have been adjusted to be more optimal but that was not done. We clarified that part in the article.
13) We remove the part concerning the evaluation of compliance by PK. This is indeed not entirely true.
We would like to thank you for your time in evaluating our work and for your constructive comments that helped us to improve the quality of our manuscript.
Best regards,

Reviewer 3 Report
There are a number of words that seem to be incorrectly used. Examples are: "efficiency' instead of "efficacy" (Abstract), "punctual" instead of "point" (Intro). Also, I am not familiar with units of "G/L"; is that the same as "x1000/uL?
The greatest concern is the limited number of subjects (n=9), with 4 of the subjects presumably being non-adherent with taking hydroxyurea as evidenced by their low MCVs. Could the low MCVs be related to alpha thal? (Were their MCVs even lower before they started taking hydroxyurea?
Author Response
Dear Editor and reviewer,
The modifications were indicated by colors to highlight changes. We would like to thank you for your time in evaluating our work and for your constructive comments that helped us to improve the quality of our manuscript.
We worked on the quality of the writing to improve its fluency and correct the mistakes.
For the units “G/l”, it is the same as 1000/µL. Both are used in our hospital routinely but we choose G/L because it is the one which is conform to international system.
We totally agree on the fact that MCV cannot be the only proof to a non-compliant patient. We collected genotypic data but only one patient had it. Therefore, we don’t know if patients with low MCV have associated alpha thalassemia. We reported MCVs before HU and the self-reported compliance to help assess adherence.
Best regards,

Round 2
Reviewer 3 Report
I think the manuscript warrants publication in JCM.